

# Enhort: a platform for deep analysis of genomic positions

Michael Menzel, Peter Koch, Stefan Glasenhardt and Andreas Gogol-Döring

MNI, Technische Hochschule Mittelhessen—University of Applied Sciences, Giessen, Hessen, Germany

## ABSTRACT

The rise of high-throughput methods in genomic research greatly expanded our knowledge about the functionality of the genome. At the same time, the amount of available genomic position data increased massively, e.g., through genome-wide profiling of protein binding, virus integration or DNA methylation. However, there is no specialized software to investigate integration site profiles of virus integration or transcription factor binding sites by correlating the sites with the diversity of available genomic annotations. Here we present Enhort, a user-friendly software tool for relating large sets of genomic positions to a variety of annotations. It functions as a statistics based genome browser, not focused on a single locus but analyzing many genomic positions simultaneously. Enhort provides comprehensive yet easy-to-use methods for statistical analysis, visualization, and the adjustment of background models according to experimental conditions and scientific questions. Enhort is publicly available online at enhort.mni.thm.de and published under GNU General Public License.

## INTRODUCTION

Some viruses like HIV (*Craigie & Bushman, 2012*) and AAV (*Deyle & Russell, 2009*) are able to copy their genomic sequence into the genome of an infected cell. This can have severe impact on host cell stability as the integration may hit and disable a gene or a regulatory region. The investigation of characteristics and underlying driving factors for virus integration is not only relevant for virology and infectious diseases research but also for approaches in gene therapy that apply virus-derived vectors and transposons to deliver functional DNA fragments into host cells (*Riviere, Dunbar & Sadelain, 2012*; *Li et al., 2015*). Each gene delivery system has its own mechanisms for genomic integration and preferences for choosing integration sites, hence different systems may have different risks for causing undesired side effects.

Next Generation Sequencing (NGS) facilitates the genome-wide profiling of integration sites, as they are collected e.g., in investigations of protein binding, virus/transposon integration or DNA methylation. Integration sites are available from databases like the Retrovirus Integration Database (*Shao et al., 2016*) and are regularly created for novel targeted vectors. Typically, the identified sites are related to a variety of genomic features and any integration preferences are determined by a comparison of actual integration sites to a set of random control sites (*Gogol-Döring et al., 2016*). A proper background

Corresponding author
Andreas Gogol-Döring,
andreas.gogol-doering@mni.thm.de

model should mimic all known biases of the signal data originating from experimental or laboratory conditions. If, for example, a profiling method is only capable of detecting integration events that are close to certain enzyme restriction sites then the control sites should also be selected accordingly.

Several tools have been published that are capable of processing genomic positions and annotations, like the Genomic HyperBrowser (*Sandve et al., 2013*). Genome browsers like the UCSC Genome Browser (*Kent et al., 2002*), IGV (*Robinson et al., 2011*) or Artemis (*Carver et al., 2011*) are designed for inspecting single genomic locations. Also custom written scripts are commonly used for the analysis of genomic positions (*Cook et al., 2014*) or libraries like PyBedTools (*Janovitz et al., 2014*; *Dale, Pedersen & Quinlan, 2011*). Once written these scripts have the benefit of being a reusable option to conduct a specific set of analysis on recurring data. However, they are limited by the available functionality because each function has be newly developed. Additionally, comparability across laboratories is afflicted by varying functionality and different implementations of background models. There is yet no specialized tool for genomic positions analysis that combines the features of instant analysis and user defined adaptable background models that mimic known biases.

In this paper we present Enhort, a user-friendly web-platform for deep analysis of large sets of genomic positions. Our aim is to accelerate and simplify the data analysis process as well as to standardize it in order to increase reproducibility. Enhort is capable of adjusting background sites used for comparison by user selected covariates. This includes annotation tracks like restriction sites or chromatin accessibility, gene expression tracks and sequence motifs. With covariates it is possible to adjust the background sites selection in a way that they match the investigated sites for a specific track. The adaptation rules out the effects of this annotation for the background. This feature can be used to adjust for experimental bias as well as specific questions. Figure 1 shows the schematic process of data gathering and the usage of Enhort in the workflow of analyzing genomic positions.

## METHODS

Integration sites of viruses are gathered by sequencing infected cells and preprocessing as shown in Fig. 1. These sites are uploaded to Enhort and are intersected with each annotation file to compute fold-change enrichment and $\chi^2$ test in comparison to control sites, yielding a measure for effect strength and significance of each annotation respectively. Figure 2 shows the schematic analysis pathway for sites uploaded by a user. Statistical analysis depends on the Apache Commons Math library (https://commons.apache.org/proper/commons-math/) and uses Bonferroni correction for multiple hypothesis testing. The libraries plotly.js (https://plot.ly/javascript/) and Circos (http://circos.ca/) are used for visualization. The results are sorted according to their relevance and presented in conjunction with appropriate figures. Example results for a virus can be seen in Fig. 3A. The software has been designed in a way that analysis results are almost immediately available after upload.

In many cases a background model consisting of random sites is not sufficient for an adequate analysis. Some protocols, for example, can only detect integration events that

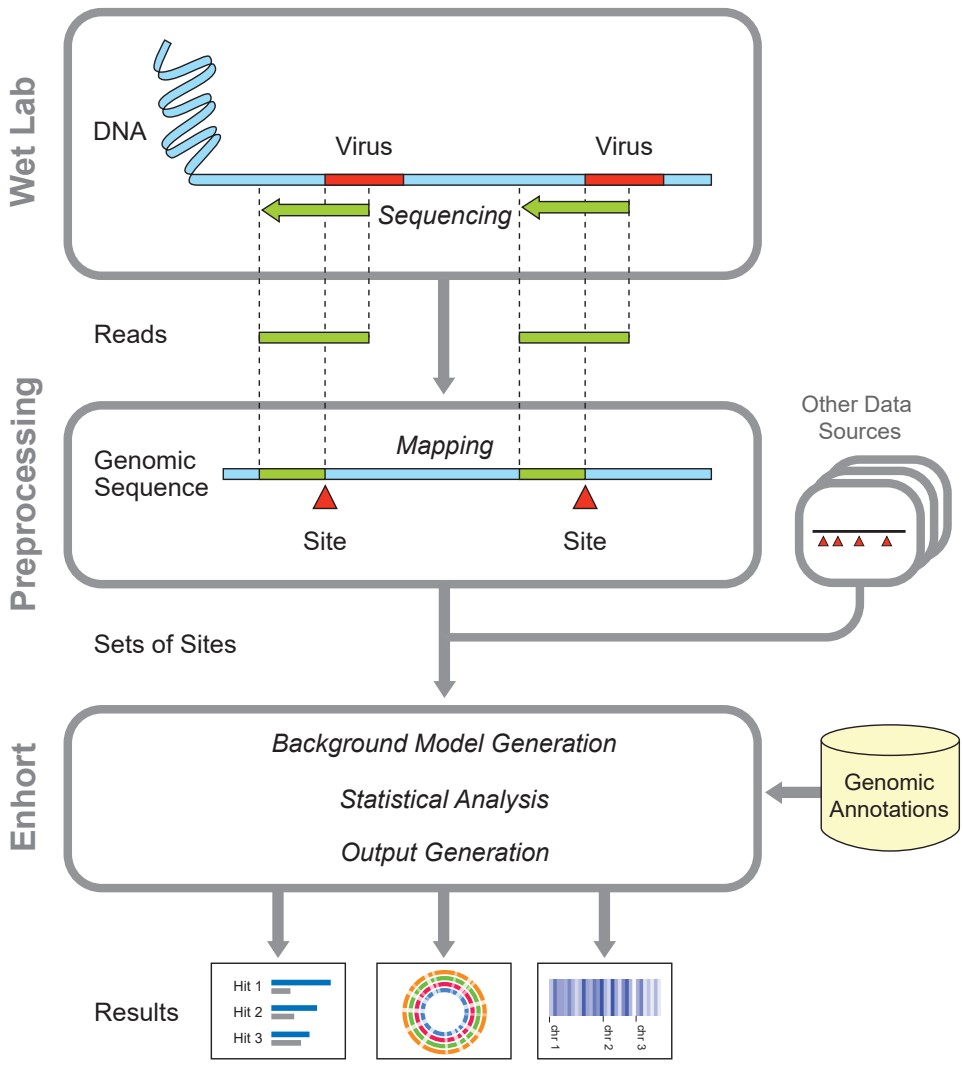

**Figure 1 Overview of preparatory work and data gathering for analysis in Enhort.** Reads containing viral integration sites are identified and sequenced in the WebLab and mapped to a reference genome. Identified insertion sites are converted to a BED file for the usage in Enhort. Together with genomic annotations from public database the analysis in Enhort is conducted to generated analysis of the given integration sites.

occurred in close proximity to a restriction site of a specific enzyme, like EcoRI, which cuts inside of GAATTC hexamers (*Pingoud & Jeltsch, 2001*). Background models should be adapted to mimic the actual integration pattern with regard to any known technical bias. In this case, the control sites should also be selected to be near restriction sites. This can be achieved in Enhort by setting the appropriate genome annotation as a covariate. When selecting the track that contains all possible genomic positions of GAATTC hexamers as covariate, Enhort will generate a set of control sites having exactly the same distribution of distances to the enzyme restriction sites as the actual virus integration sites.

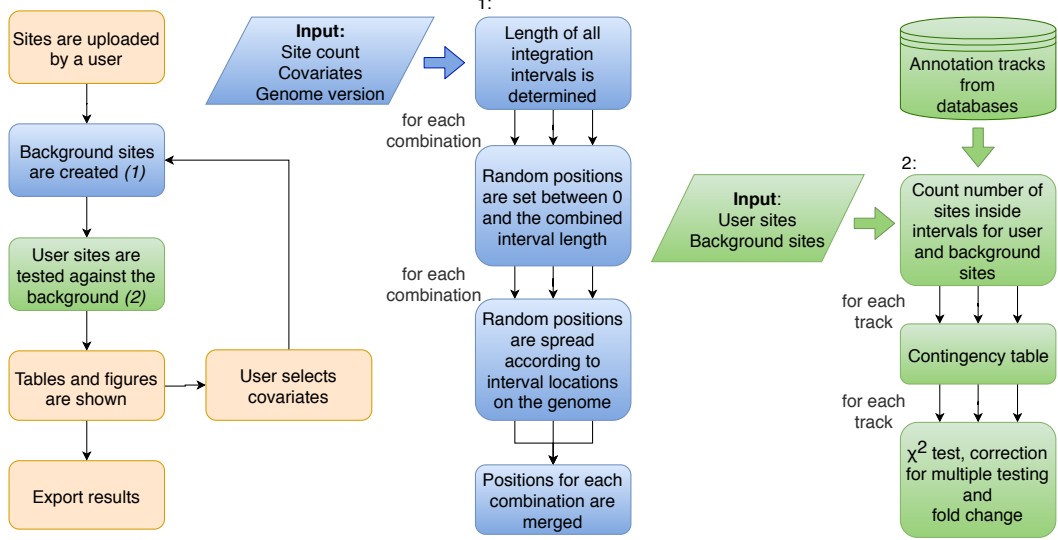

**Figure 2 Flowchart of the procedure of analysis performed by Enhort.** Blue boxes show the steps to create a background model based on multiple covariates. Random positions have to be set for each combination of covariates. Green boxes show the steps to test the user sites against the background sites. The results are returned as a table and converted into figures for the user.

Covariates help to adapt the background model both for technical circumstances, for example, restriction sites and for eliminating a bias or biological preferences such as motifs or genetic features. Covariates can also be used to identify dependent or separate weak integration preferences that are covered by stronger effects, as shown in Fig. 3B. MLV integration sites are compared to two different control sets: A random and an altered background, to identify the actual integration preferences; e.g., for histone mark H3K4me3, which is a known preference of MLV (*Gogol-Döring et al., 2016*).

For the validity of statistical testing it is usually indispensable to normalize the background model relative to multiple covariates. For that purpose, Enhort supports the selection of multiple covariates simultaneously in order to further investigate the integration site characteristics. For example, Enhort may create a control set that considers chromatin accessibility, restriction site distance as well as several histone modifications simultaneously. This functionality is needed to build background models for sites that are influenced by multiple factors, e.g., biological and technical biases. A set of additional features listed in the following table:

1. Statistical analysis for annotation tracks:

    (a) Fold change
    (b) $\chi^2$ test
    (c) Kolmogorov–Smirnov test

2. Hotspot analysis (Fig. 4C)
3. Position depended enrichment (Fig. 4A)
4. Background models based on:

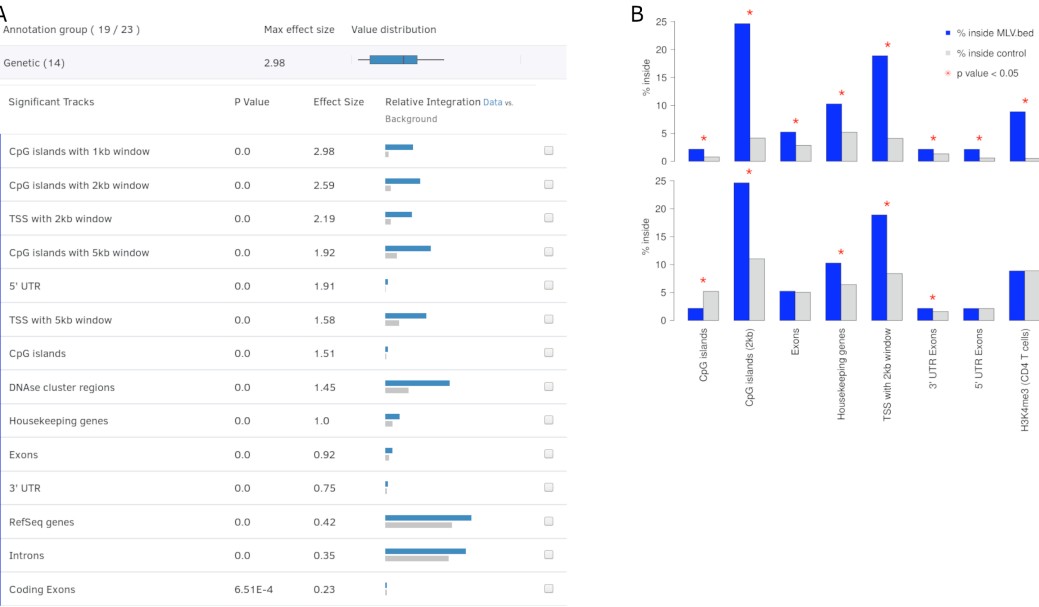

**Figure 3** **Output view example, generated by Enhort when analyzing Murine Leukemia Virus (MLV) integration sites in CD4 $^+$ T cells** (*Roth, Malani & Bushman, 2011*). (A) The results are presented in a table containing for each annotation the *p* value, effect size and a visual representation of the integration. The annotations are ranked by effect strength. (B) Effect of covariate selection. The upper diagram contains integration frequencies of MLV compared to random sites for a selection of annotations. This virus is known for preferentially integrating near transcription start sites (TSS) and H3K4me3 histone marks (*LaFave et al., 2014*). The lower diagram shows the same data after selecting H3K4me3 as covariate. The adapted background model is generated in a way that control sites and MLV integration sites have the same frequency relative to H3K4me3. This also changed the control site frequencies for other annotations: MLV integration is no longer enriched but depleted in CpG islands when compared to the adapted background model.

    (a)   Inside and outside of annotations
    (b)   Distance to annotations
    (c)   Scored annotations
    (d)   Sequence logo

5.  Upload background sites
6.  Comparing effects of different background models
7.  Batch analysis of multiple integration sets
8.  Heatmaps to compare integration sets (Fig. 4B)
9.  Custom annotation tracks
10.  Blend annotation tracks
11.  Export results as R code and CSV files

    Enhort is separated into a lightweight, web-based user interface and a high performance back-end server attached to a SQLite database storing meta-information about the annotations fetched from DeepBlue (*Albrecht et al., 2016*). Results from Enhort are instantaneously available as seen in Table 1 where the run times for different input sizes are shown. Our application currently offers 1402 annotation tracks from 97 cell

**Table 1** **Analysis execution times for different usual site counts, annotation tracks from hg19 and co-variate counts.** (Back-end server: SuperMicro SuperServer 4048B-TRFT 4x Intel Xeon E7-8867v3 with 2048GB DDR3 ECC LR).

| Track count | | 23 | | | 1,127 | |
| --- | --- | --- | --- | --- | --- | --- |
| Covariate count | 0 | 2 | 5 | 0 | 2 | 5 |
| Site count | | | Execution time (ms) | | | |
| 150k | 877 | 1,188 | 4,668 | 8,538 | 10,436 | 12,540 |
| 125k | 717 | 1,103 | 5,628 | 5,509 | 7,552 | 7,975 |
| 100k | 749 | 817 | 5,085 | 3,724 | 4,672 | 8,673 |
| 75k | 624 | 571 | 4,019 | 4,905 | 4,397 | 9,633 |
| 50k | 470 | 555 | 5,455 | 4,736 | 5,844 | 10,451 |
| 25k | 308 | 351 | 4,628 | 3,246 | 3,091 | 8,111 |

lines and tissues for human genome assemblies hg19 and hg38, downloaded from UCSC Genome Browser (*Fujita et al., 2011*), Encode (*ENCODE Project Consortium, 2004*), ChIP-Atlas (http://chip-atlas.org), BLUEPRINT Epigenome (*Adams et al., 2012*) and Roadmap Epigenomics (*Roadmap Epigenomics Consortium et al., 2015*) using the DeepBlue Epigenomic Data Server (*Albrecht et al., 2016*).

# RESULTS AND DISCUSSION

## Literature review

We reviewed the relevance of Enhort for contemporary research by systematically searching PubMed, Google Scholar, and several review articles for publications concerning the analysis of genomic integration sites. The publications include virus integration site analysis for HIV, MLV, HRP-2, SIV, foamy virus, HPV, AAV and transposons such as piggyBac, LINE-1, Alu and sleeping beauty. In total we identified 59 relevant publications. Details on the reviewed publications and methodological analysis are available in the Table S1. Of these publications 19 used completely random control sites, only six used adapted control sites. The data analyses presented in 37 (63%) publications could have been entirely performed with our tool. Six further publications use at least some methods provided by Enhort. We assume that if they had the opportunity to use Enhort the authors would have saved a lot of effort writing custom analysis scripts.

## Data re-analysis

To further present the capabilities of Enhort we re-analyzed integration sites of the PiggyBac transposon (PB) published by *Gogol-Döring et al. (2016)* using Enhort. Results from *Wilson, Coates & George (2007)* are used for comparison. PB integration characteristics show a preference for genes, exons, introns, highly expressed genes, DNase I hypersensitive sites, H3K4me3 and open chromatin structures (*Wilson, Coates & George, 2007*; *Li et al., 2013*). We uploaded the PB integration sites to Enhort, selected all relevant tracks and finally exported the results. Figure 5A shows the log fold changes for a selection of annotations for PB against a random background in grey. Figure 5B shows the sequence logos for the

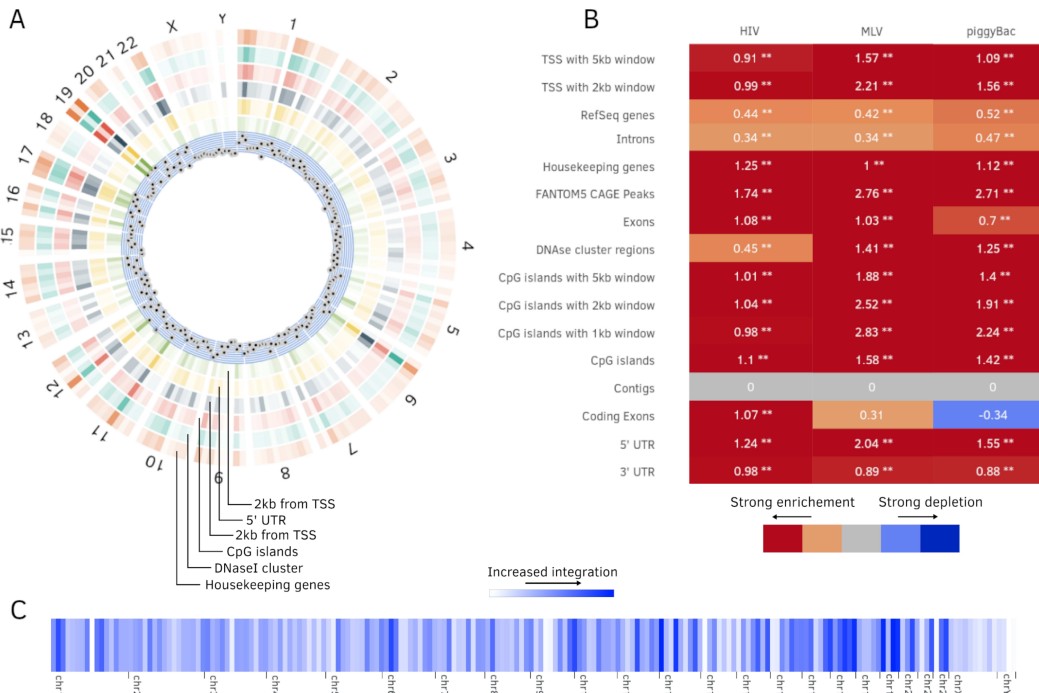

**Figure 4   Additional plots generated by Enhort.** (A) Circos plot (*Krzywinski et al., 2009*) of position dependent enrichment over all chromosomes for MLV for the most significant tracks. (B) Heatmap for a set of three integration data sets against various annotations. The values are $\log_2$-fold changes of the numbers of integration vs control sites falling into a given annotation. Star symbols mark statistically significant changes. The same background sites are used for the comparisons. The background sites are adapted to integrate only inside the sequence contigs. (C) Integration hotspots across the genome for MLV. The color intensity of the thin bars show the integration ratio inside of the respective genomic region.

PB integration sites and the random background. The barplots were created using the R-export feature of Enhort.

The key feature of the PB integration preference is the TTAA motif in which all integrations occur. To precisely analyze the preferences of PB integration the background model has to be adapted to replicate the TTAA motif preference. This can be achieved using Enhort by creating a set of pseudo-random control sites that are located only inside a TTAA sequence. To achieve this, we simply selected the sequence logo as a covariates. Enhort takes genomic positions from a pre-sampled set of positions where each position has a probability based on the similarity between the surrounding sequence and the TTAA sequence. The results are shown in Fig. 5C where the background sites and PB show a similar motif after the motif is added as a covariate using Enhort. The motif adaption also changes the observed integration characteristics seen in Fig. 5A. The relative decreased integration of PB into coding exons is changed to a significant preference, because CpG islands are less likely to be hit by a site from the adapted background model, as TTAA occurs relatively less frequent in CpG islands. The same applies to DNAse cluster regions, TSS and exons, where the significance of integration is enriched in comparison to a random background. Only a small change for the enrichment in introns and genes is visible. Overall

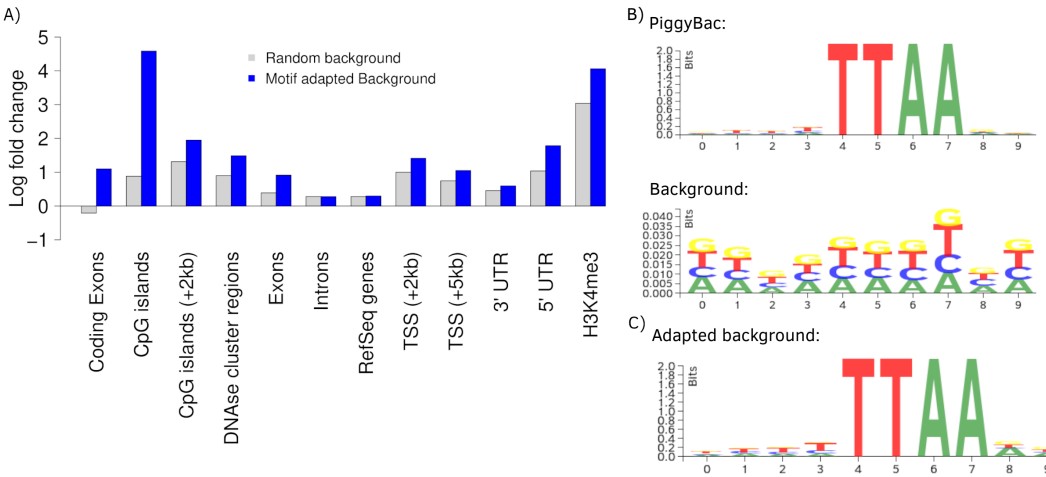

**Figure 5** **Analysis of PB integration sites.** (A) Log fold changes of PB integration sites in relation to several annotations against a random and an adapted background model. Changing the background model to adapt the TTAA motif changes the observation of several integration preferences. (B) The PB motif and random sites motif, corresponding with the random background bars in (A). (C) Motif of the random sites after adaption to the PB motif using Enhort.

**Table 2** **Log fold changes and integration ratios of *Wilson, Coates & George (2007)* in comparison to Enhort for two PB integration site sets.**

| Annotation track | Enhort Fold change | Wilson et al. Fold change | Enhort PB (%) | Wilson et al. PB (%) | Enhort Random (%) | Wilson et al. Random (%) |
|---|---|---|---|---|---|---|
| RefSeq genes | 1.32 | 1.46 | 63.08 | 48.8 | 47.93 | 33.2 |
| TSS (±5 kb) | 2.14 | 3.00 | 20.8 | 16.2 | 9.7 | 5.4 |
| CpG islands (±1 kb) | 5.52 | 2.00 | 12.99 | 3.8 | 2.35 | 1.9 |
| CpG islands (±5 kb) | 2.82 | 0.96 | 22.85 | 7.7 | 8.09 | 8.3 |
| Repeats: | | | | | | |
| LINE | 0.71 | 0.76 | 7.72 | 12.7 | 10.90 | 16.7 |
| SINE | 0.50 | 0.54 | 3.8 | 6.0 | 7.64 | 11.1 |
| LTR | 0.56 | 1.84 | 2.79 | 6.8 | 5.0 | 3.7 |
| DNA | 1.61 | 1.18 | 1.87 | 4.0 | 1.61 | 3.4 |

this indicates that beside the TTAA preference of PB there are additional mechanisms that alter the integration preferences. Using the background adaption feature of Enhort it would be possible to test different hypothesis against the data and build a model that explains the integration preferences.

To further review the analytic capabilities of our software, the integration counts of PB sites are compared to published results from *Wilson, Coates & George (2007)*. The comparison can be seen in Table 2. An increased integration of PB into RefSeq genes, inside the 5kb-TSS window, as well as a preference for CpG islands is observable for both analyses.

*Wu et al. (2003)* published a study on MLV and HIV stating that MLV favors TSS regions, whereas HIV does not display a strong preference towards TSS regions. The

**Table 3   Comparison between fold changes of *Wu et al. (2003)* and Enhort over different annotations on the same integration sites.**

| | Wu et al. | | Enhort | | | | | |
|---|---|---|---|---|---|---|---|---|
| | **HIV** | **MLV** | **HIV** | **MLV** | **HIV**[a] | **MLV**[a] | **HIV**[b] | **MLV**[b] |
| RefSeq genes | 2.58* | 1.5* | 1.7* | 1.4* | 1 | 1 | 1 | 1 |
| Housekeeping genes | – | – | 3.7 * | 1.36 | 2.22* | 1.12 | 2.05* | 1.04 |
| CpG islands (±1 kb) | 1 | 8* | 0.41 | 6.24* | 0.35 | 6.17* | 0.31 | 4.09* |
| TSS (±5 kb) | 2.5* | 4.7* | 1.34 | 2.3* | 1.14 | 2.02* | 1 | 1 |
| H4K20me1 | – | – | 1.71* | 1.56* | 1.34* | 1.52* | 1.36* | 1.42* |
| H3K4me2 | – | – | 1.23 | 21.7* | 1.48 | 21.29* | 1.09 | 15.2* |
| H3K27ac | – | – | 0.9 | 24.52* | 1.01 | 22.79* | 0.83 | 20.12* |

**Notes.**
*$P < 0.002$.
[a] with RefSeq genes as covariate.
[b] with RefSeq genes and TSS (± 5 kb) as covariates.

available integration sites were uploaded to Enhort and analyzed using the batch tool with a random 10,000 site background model. The results from Enhort show a similar integration pattern as stated in *Wu et al. (2003)* (Table 3). Except for CpG islands for HIV where Wu et al. found a near random integration and we found a decreased integration.

For further review, HIV and MLV integration sites were uploaded independently to Enhort, and RefSeq genes added as covariate. This background model had only a little effect on MLV as the preference for TSS and CpG islands only changed slightly, indicating that the preference for TSS is not due to a preference for RefSeq genes. For the HIV integration sites the housekeeping genes, which are a known preference of HIV (*Craigie & Bushman, 2012*), are still statistically significant against this background model.

Finally, RefSeq genes and TSS (±5 kb) were both used as covariates together, showing that the integration ratio of MLV into CpG islands with a (±1 kb) window decreases slightly. This shows that the integration into the CpG islands is probably not a side effect of the preference for TSS or genes. The combined background model with RefSeq genes and TSS does not have any influence on the HIV fold changes compared to the previous background model.

The creation of each background model and comparing the results was possible using built-in features of Enhort. We further added histone modifications to the analysis showing that H4K20me1 is significantly enriched for both integration sets and does not change significantly for the different background models. This indicates that the histone modification preferences is an additional effect, only slightly influenced by the preference for genes and TSS. H3K4me2 and H3K27ac are known preferences of MLV (*De Ravin et al., 2014*) and show a high fold change for all background models. With the available database it would be easy to add numerous additional annotations for comparison.

We have shown that Enhort is capable of reproducing integration site analysis with less effort and additionally offers easy-to-use mechanisms to create more sophisticated analysis using adaptable background models. The exact annotation files were not available for comparison, so it was not possible to produce the exact numbers. However, Enhort uses

the same calculation principle. With the same annotations and sites the results by Enhort would be the same as in the referenced publications.

## CONCLUSION

In this publication we present Enhort, a fast and easy-to-use analyzing platform for genomic positions. Based on a comprehensive library of genomic annotations, Enhort provides a wide range of methods to analyze large sets of sites. In contrast to multi-purpose software such as bioconductor, Enhort enables scientists to analyze data without programming effort or extensive manual work.

Our literature review shows that Enhort is able to perform most of the analyses commonly used in the investigation of integration sites. The re-analysis of *Wilson, Coates & George (2007)* and *Wu et al. (2003)* demonstrates that Enhort is able to reproduce analyses from literature with little effort. It was not possible to reproduce the exact values, because the version of the annotation was not recorded in the publications. However, more detailed insights can be made using adaptable background models. This was shown in the comparison of HIV and MLV from Wu et al. against different control sites.

Most publications use very simple background models for statistical analysis of integration data and could potentially be improved using better background models. Enhort provides methods to easily create more sophisticated background models for improving both the accuracy and the range of possible analyses. Complex background models can be used to identify weak effects and segregate driving factors for integration, find a minimal set of annotations to mimic integration characteristics, as well as to eliminate technical biases. In conclusion, this shows that Enhort will be a valuable tool for further analyses of genomic positions, no matter if these positions are derived from virus integration, sequence motifs, enzyme restrictions, histone modifications, or protein binding.

### Funding

This work was supported by the Hessen State Ministry for Higher Education, Research and the Arts. The funders had no role in study design, data collection and analysis, decision to publish, or preparation of the manuscript.

### Grant Disclosures

The following grant information was disclosed by the authors:
Hessen State Ministry for Higher Education, Research and the Arts.

### Competing Interests

The authors declare there are no competing interests.

### Author Contributions

- Michael Menzel conceived and designed the experiments, performed the experiments, analyzed the data, contributed reagents/materials/analysis tools, prepared figures and/or

tables, performed the computation work, authored or reviewed drafts of the paper, approved the final draft.

- Peter Koch contributed reagents/materials/analysis tools, performed the computation work.
- Stefan Glasenhardt contributed reagents/materials/analysis tools, performed the computation work.
- Andreas Gogol-Döring prepared figures and/or tables, authored or reviewed drafts of the paper, approved the final draft.

## Data Availability

The source code and build instructions are available at https://git.thm.de/mmnz21/Enhort.

## Supplemental Information

Supplemental information for this article can be found online at http://dx.doi.org/10.7717/peerj-cs.198#supplemental-information.

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
