# Peer review of "Enhort: a platform for deep analysis of genomic positions"

_PeerJ Computer Science, doi:10.7717/peerj-cs.198_

## Round 0.1 · original submission · Major Revisions

Firstly, please accept my apologies in the delay in returning these comments to you.

In general, both reviewers found your submission interesting and relevant, but raised substantial points that require clarification.

1. Applicability. Enhort is presented in paragraph 1 as a tool specifically design to perform statistical analysis to identify genomic markers that contribute specificity/bias in viral integration, but little further detail is provided.

i. Generality of analysis. The kind of probabilistic sequence motif/annotation cross-correlation analysis that Enhort performs would seem to be useful in other contexts (e.g. exploration of RNASeq fold changes w.r.t. sequence motifs, etc). A better description of the statistical analyses that Enhort performs would help readers evaluate this themselves (See point 3 below), but you could also suggest these alternative use cases in your discussion/conclusion.

ii. Precisely describe sources of experimental data. in paragraph 2 you simply state that 'Next generation sequencing' facilitates observing these sites. Please mention and cite specific kinds of sequencing experiment that can generate the kinds of data amenable to analysis by enhort (e.g. do you simply mean 'sequencing a whole organism' or are more specific methodologies required such as crosslinking, chip-seq, or another kind of library selection approach). Since this is a PeerJ-CS submission, you should also at least briefly outline the typical data processing pipeline required in order to generate data amenable for analysis using Enhort.

2. Mention related work/other tools and properly compare Enhort's capabilities. R2 points out that a number of tools that exist that either provide positional analysis of genomic data (e.g. libraries in bioconductor/R), or visualisation of positional annotation (such as analysis results) on genomic coordinates (IGB, IGV, and others). These should at least be mentioned in your introduction, and a formal comparison (however brief) included in your discussion to examine how their respective capabilities measure up to those provided by Enhort. In particular, your anecdotal comments regarding Enhort's potential for labour saving could be more thoroughly justified if you provided a more thorough explanation of Enhort's analysis and visualisation procedure, and what steps would be required in order to achieve the same results without Enhort.

3. More detailed description of Enhort's analysis methodology.
i. Methods section needs to more clearly define inputs to analysis, and properly reference Enhort's statistical methodologies, and any implementations it depends on (e.g R packages, etc).
ii. Ideally you should cite a publication where the efficacy of this formulation is analysed in more detail with respect to its utility for answering these kinds of biological questions (for instance, Gogol-Doring 2016 is not sufficiently detailed for this purpose).
iii. R2 noted that Enhort's 'interactive background choice' capability is potentially of great interest, but needs to more clearly explained. R1 requests more detail be provided about the statistical validity of Enhort's approach, since it could potentially result in erroneous conclusions unless its methodology includes corrections for multiple hypothesis testing. Please ensure these requests/concerns are addressed.

4. 'Literature Review'. Both reviewers are unhappy with your validation of Enhort's utility by 1) inspection of published studies in the literature and 2) demonstrating that Enhort can reproduce results from just one existing study (which was originally performed by one of the authors).
i. Neither consider your literature analysis rigorous, and request you present more detail to support your claim. R1 alternatively suggests you rephrase this section as an anecdotal 'justification' section within the discussion. In general, If you wish to present such an analysis, please treat it as a qualitative or semi-quantitative literature survey: ie. you must at least include a formal description of how you identified and evaluated each study, and provide details (perhaps as supplementary information) on how each paper identified measures up to your criteria.
ii. Both reviewers suggest presenting an evaluation of Enhort's results for data from other studies. This may prove to be technically difficult; so, providing a more detailed presentation and evaluation of Enhort's methodology is included in the manuscript, inclusion of additional case studies may not be necessary. Performing these additional analyses, however, could have other benefits (see 5.ii below), so please consider if it is feasible to do so.

5. Software and web site.
i. availability and installation documentation. R1 notes that the website was unavailable - presumably because it is at enhort.mni.thm.de - this must be fixed in the manuscript. R1 also notes the source distribution was not easy to build. As GPL licensed software it is important that others are able to build and deploy Enhort, and it would help to provide the sequence of commands needed to build and deploy all components on OSX or Linux, or access to tarballs of prebuilt binaries (I recommend you provide instructions for OSX based on the use of conda/bioconda as primary package manager).
ii. Scalability and performance. R2 notes that SQlite is unlikely to be sufficient for large scale analyses - particularly since the tool is designed as an interactive web application that could be subject to heavy loads. Inclusion of a statement or table in the manuscript to report performance of the Enhort platform with different size datasets (e.g. some selected from your literature survey or failing that, realistic synthetic datasets) would help support your response here.

6. Grammar/Typos. R1 helpfully details a number of specific typographic issues in their 'Basic Reporting' section.

·

Basic reporting

In "Enhort: a platform for deep analysis of genomic positions" Menzel et al present a new tool aimed at the investigation of viral integration sites in the human genome. The main addition to the state of the art is the user-friendly method for incorporating prior knowledge (i.e. biological or technical biases) in the analysis of these sites.

**English and language**: Overall, the paper is written in coherent and legible English, albeit that it includes some typos and the odd unfinished sentence. These include (but might not be limited to):
* l55: to control sites. Yielding a measure => to control sites, yielding a measure
* Legend figure 2: integration vs. controll sites => integration vs control sites
* l76: distance as well as for several histone => distance as well as several histone
* l78: Without background models build for multiple covariates => Without background models built for multipe covariates
* l80-82: This is not a correct sentence (including a non-closure of the parentheses on l82).
* l116: Figure 3B shows the screenshots of sequence logos => Figure 3B shows the sequence logos
* l119: To precisely analyzes the preferences => To precisely analyze the preferences
* l119-120: to be adapted to replicated the TTAA motif => to be adapted to replicate the TTAA motif
* l124: characteristics seen Figure 3A => characteristics seen in Figure 3A
* l125: significant preferences => significant preference
* l132: To further review the analytic capabilities the integration counts => To further review the analytic capabilities, the integration counts

**Structure**: The structure of the manuscript reflects the contents and best practices. However, the "Discussion" section does not contain any discussion. I suggest renaming the "Results" section to "Results and Discussion", and the "Discussion" section to "Conclusion". Similarly, the section "Literature Review" should be renamed to "Possible impact of Enhort" or similar.

**Definition of terms etc**: Although most terms are well-defined, it is not clear in the legend of figure 2 what a contig refers to ("contig" in the genome assembly sense?)

Experimental design

It would have been good to try the tool out, but unfortunately the website cannot be reached. Similarly, the gitlab repository does include build instructions as well as an example database, but this is insufficient for review as I encountered issues installing maven on my computer which is necessary to try the tool out.
Having the website running is a sine-qua-non before this manuscript can be accepted. In addition, it would be good if the authors provided e.g. a docker image with example data which would also include a walkthrough of the patterns that one is looking for.

Validity of the findings

As the user can try out adding different biases to the background model, the authors should take into consideration the effect of multiple testing: is this relevant here or not, and why. See e.g. l129-130: "Using the background adaption feature of Enhort it would be possible to test different hypothesis"

The last sentence of section "Literature Review" states "Since data analysis only takes minutes using Enhort, we assume that the authors of these papers would have saved a lot of effort if they had used our tool." This is obvious speculation and should be identified as such. Were the authors of those papers contacted to try out Enhort?

Additional comments

In addition to the structured comments above, there are some additional points to be made:

[1] On l122-123, the authors mention that the background sites in 3C and those for PB show a similar motif after the motif is added as a covariate using Enhort. Isn't this a logical consequence? If you'd measure lengths of individuals and the average thereof is higher than those for women, you could similarly add gender as a covariate and of course your average would become identical with that for women...
[2] The authors state in l145-146 that "The analysis of PB integration sites demonstrates that Enhort is capable to reproduce analysis from literature with little effort." However, "from literature" refers to a single published paper (i.c. Wilson et al 2007). There are 2 issues with this. First, being only compared to a single paper does not constitute proof that a method can actually be applied more generically, and can almost be interpreted as anectodal evidence. Second, in comparing Enhort with Wilson et al, the authors had the advantage of hindsight and knew what they were looking for. Therefore, reproducing results "with little effort" becomes less meaningfull. I suggest the authors include a comparison with at least one other study, and take consideration of the fact that they already know what they're looking for in this comparison. A way to circumvent this problem is to make predictions using Enhort that are later validated in a separate study.
[3] It is not completely clear what the authors mean at l116 with "sequence logos for PB". Are these sequence logos of known integration sites?
[4] Figure 2 misses legends for all 3 subfigures: what are the different bands in 2A? What is the colour scale in 2B? Is 2C just a linear version of 2A?

Reviewer 2 ·

Basic reporting

This manuscript describes Enhort, a system for associating annotation and genomic position data. The system relies on existing statistical methods to provide familiar statistics and visualizations that would support a broad variety of analyses.

The high-level organization of the manuscript is reasonable. The figures are high-quality and informative. The writing at the beginning of the manuscript is relatively high quality; however, as the manuscript progresses, the writing becomes less clear and contains more typos. The manuscript would benefit from an end-to-end editing pass.

The manuscript contains a number of useful references to example studies that Enhort would support. These references are helpful for understanding and contextualizing the motivation for the system. However, the specific contributions of the system beyond existing approaches is unclear in large part due to a lack of comparison to existing systems. The manuscript notes that many systems exist for simultaneous position and annotation analysis, but only one such system is noted. The claimed difference appears to be that this system provides "instant analysis and user-defined adaptable background models that mimic known biases." Without a comparison to existing approaches, it is difficult to see how Enhort truly achieves this goal. Systems like IGV, Artimis, and Bioconductor all enable annotation analysis alongside genomic position and range data and appear to share many common features with Enhort. A broader discussion of the contribution of the system in relation to this work would significantly strengthen the manuscript.

Experimental design

There are a number of missing details as to the actual functionality of the system and how the system achieves the technical contributions noted above and extended on in the introduction as to "accelerate and simplify the data analysis process as well as to standardize it in order to increase reproducibility." The primary method to achieve this in Enhort is through the integration of user-selected covariates to serve as a control site. The idea of integrating covariates to serve as controls in annotation analysis is intriguing. However, there is little discussion of how these covariates are integrated, how the system might support the selection of relevant covariates, or how the system supports transparency in how the covariates shift analysis after integration. A use case illustrating the target analysis workflow and how Enhort manages this workflow would be helpful in illustrating these methods as would a formal comparison to additional approaches (see above).

The deployment of Enhort as a web-based application seems like a useful method for deploying and maximizing the impact of the system. The system appears to include access to several existing samples. However, given the aim stated in the abstract of targeting large-scale datasets, the choice of a SQLite backend seems like it may be limiting. A performance analysis or discussion of scalability would be useful and such a discussion should tie to the impact of scale on the integration of covariates as well as how covariate integration might influence performance.

The system appears to integrate a number of standard analytical tools for statistical and visual analysis, which, coupled with the web-based implementation, provides a useful and accessible method to integrate familiar analyses. However, these tools are described as if they are only used once data has been fully integrated into the system. It is unclear how the system supports either the instant integration or standardization that are noted as target contributions. Additional discussion of what exactly is intended in these contributions as well as how Enhort is architected to support these factors should be included.

Validity of the findings

Enhort is evaluated primarily using a literature review of prior analyses. However, the conclusion of this review is that a large percentage of the target analyses could have been completed using Enhort, and the manuscript speculates that the analyses could have been conducted more efficiently with the system. The speculation is backed by a specific case study attempting to reproduce Gogol-Doring et al's 2016 results and Wilson et al's PB site analyses.

This method of evaluating the system seems limited given the lack of comparison against existing approaches. For example, what grounds are there for the speculation that Enhort could have saved substantive effort in the prior papers? How does Enhort's methods improve on these prior analyses? What novel components of Enhort allow for these analyses to be conducted in ways that prior methods could not achieve?

The case study evaluation is also somewhat limited. While the case study does surface alternative hypotheses for the 2016 data, no discussion is provided as to how comparable the analysis is to the original approach. Further, the comparison to the 2007 data is limited: was there other data or testing that ruled out these hypotheses or are these hypotheses from the 2007 study that required an approach like that offered by Enhort to achieve? While a lack of full replicability on an older study is understandable, it also weakens the argument that Enhort is superior to existing methods. It may be more interesting to see cases where Enhort could improve on existing methods or at least to provide some evidence showing how the tool significantly aids biological analysis workflows.

Overall, the validity of the claims in the paper is not well-supported by the current evaluation. Specific cases illustrating where Enhort can improve on current approaches or offer novel insights into data as a result of its covariate model and other features would provide stronger evidence of the validity of the system's contributions. The evaluation should also more directly substantiate the specific claimed contributions of the method, notably those of instant, accelerated, and simplified analyses. The current evaluation makes efforts towards the replication claims; however, it should also be substantially strengthened either through a deeper discussion or through collaborations with scientists

Additional comments

The manuscript overall would benefit from a stronger focus on the specific technical contributions of the system. This includes a comparison to existing approaches to highlight the novelty of the approach, a discussion of how the system achieves these novel aims, and an evaluation more directly illustrating the claimed contributions. Enhort appears to bundle interesting statistical approaches with thoughtful considerations of necessary analysis components; however, without a clearer discussion and evidence of the utility of these contributions, it is difficult to assess the novelty and impact of the system.

---

## Round 0.2 · Minor Revisions

Thank you for your revised manuscript. Reviewer 1 and I agree that in general, our major concerns have been addressed and the manuscript is significantly improved. There are, however, several minor points that remain.

Below I itemise specific issues raised by Reviewer 1, and offer options for addressing them:

1. The claim that Enhort provides a way to perform these analyses more quickly than those implemented to compute statistics for studies found in the literature (such as those identified in the literature review presented in the results section).

i. The introduction could be expanded to more clearly describe existing implementation approaches (e.g. how many used ad hoc R scripts, or a specific library).

ii. The results section could summarise how often each methodology highlighted in the introduction was encountered to report whether there is already 're-use' of existing methods (and if those methods are already the same as those Enhort provides access to, all the better !).

iii. R1 suggests that if no evidence can be presented, you should withdraw claims about Enhort being 'less time consuming' (line 117-118). Whilst I do not consider it essential that you consider conducting a robust study regarding the usability of Enhort with respect to general purpose scripting tools, more detail will help support your claims in this regard, and it is not appropriate to make these claims otherwise.

2. Explicitly list and describe the visualisations and analysis capabilities that Enhort provides.
i. Here a brief list in the text is probably sufficient, but a table (as requested by R1) would be satisfactory.

3. Enhort's results differ from those reported by Wu et al. - this discrepancy warrants at least a brief justification.

i. You already mention in your rebuttal that you avoided any speculation in this regard but you should at least highlight in your conclusions that you have demonstrated the potential efficacy of Enhort in facilitating a more 'nuanced' (ie more realistic) re-analysis of integration site observations.

4. Benchmark platform details.
i Please provide details of hardware platform used when reporting timings/performance on benchmark datasets.

5. R1 highlights layout issues in Figure 1 and 2 that should be addressed.

Additionally, please find my own specific requests, and finally some typographic revisions.

6. In line with R1's request (item 1 above), additional context in still required to introduce how Enhort is used in practice. e.g. the methods section starts at L. 56 with the statement 'Uploaded integration sites..', but at no point earlier in the text have you actually explained how integration sites are provided.

i. At L.56 please refer the reader to Figure 1 (which does explain the context when one might begin to use Enhort).

ii. Figure 1 requires a brief legend to explain in words the data processing steps outlined pictorially (e.g. Reads containing viral integration sites are identified by mapping to the host's reference genome and detection of positions where viral genomic sequences have been inserted. Identified insertion sites in the host genome are converted to a set of BED annotaton lines for import to Enhort. ..'

7. L184-185. Since this paper does not provide categorical evidence that simple background models result in false or poor quality integration site statistical analysis, you cannot really make such a strong statement.

i. Notwithstanding any additional text included in response to point 3 above highlighted by reviewer 1 that may modify this statement, I recommend that you simply join the two sentences to allow for reasonable doubt:
'Most publications use very simple background models for statistical analysis of integration data, and could potentially be improved by using better background models.'.

8. Minor/Typographic revisions

L. 13. virus integration or DNA methylation. 'and' not 'or'

L. 52. 'This feature that can be used' - remove 'that'.

L. 75. 'Covariates are also usable to identify'. Sugg. 'Covariates can also be used to identify'.

L76. 'as it is shown' - remove 'it is'.

L76-78: proper punctuation needed:
'MLV integration sites are compared to two different control sets: a random and an altered background, to identify the actual integration preferences; eg. '

L85: 'comprehensive' appears twice - suggest you simply omit the first occurence.
L93: 'for plotting and usable for publication' - change to 'for plotting and publication'.

L97. 'also used as covariate' - should be 'a covariate'

L113. Please mention that the 59 relevant publications are listed in the supplemmentary table, along with the methodological analysis presented in the next few lines.

L138. suggest revised to 'adapted background model, as TTAA occurs relatively less frequently in CpG islands'.

L153. Insert commas : 'For further review, HIV '...' independently to Enhort, and RefSeq'

L162. 'TSS does not has any' - change to 'TSS does not have any'.

L167. 'only influenced minor by' - bad grammar - suggest 'and only slightly influenced by'.

L170. 'effortless' - a more usual form of the phrasing here would use 'easy'.

L171. 'It can be shown that' - this is not really a fitting phrase at the *end* of the results section. You probably mean to say - 'We have shown that Enhort is capable of'.

L181. 'is capable to' - change to 'is able to'

L182. split to two sentences - '... be made using adaptable background models. This was shown ...'.

9. References

L202,L204 - duplicate reference (Cragie 2012 a and b)

Several references appear to have their title lowercased (e.g. the ENCODE project canonical paper). See L211, L226, L239, L266, L272

Reviewer 2 ·

Basic reporting

The overall language in the paper has been significantly improved. Remaining issues include:
* Shao et al. is noted outside of the parentheses
* "symbols mark statistical significant"--statistically
* "vs controll sites" -- control
* "are used as comparison" -- as a comparison (or for comparison)
* "enables scientist to" -- "enables scientists to"

The added figures help clarify the workflow. The figures could be improved in the following ways:
* Figure 1: The Enhort cell's text is not centered.
* Figure 2: The fact that two charts on the right expand on steps on the leftmost chart is not obvious, especially given the spatial overlap between the blue chart and primary chart. Labels, call-outs, and/or improved spacing could help clarify that the blue and green charts are subsets of the original flowchart on the left. Further, the text spacing relative to both the background cells and between cells (e.g., "for each combination") suffers from significant visual tension (e.g., many of the cells are too small so the text is close to the cell borders) and should be revised.

Experimental design

The response letter was helpful in understanding the primary contribution and importance of the approach (e.g., providing an automated WYSIWYG system for conducting the target set of analyses); however, these details are still largely missing from the paper (they do not appear until the conclusion). The Introduction should make clear what the current workflows look like (e.g., custom scripts ideally with citations to example studies), the limitation of current approaches, and the workflows that Enhort supports to overcome these limitations. Further, custom scripts represent a trade-off: they require time to develop, but are flexible, often highly reusable and can work with custom data formats. It would be useful to have some discussion of the trade-offs of Enhort and scripts constructed using existing packages.

The Methods section is also significantly improved; however, there is still a limited sense of what specific features Enhort offers. For example, what visualizations are included? How are the "appropriate figures" selected? What kinds of additional statistical and analytical support is provided? The Methods discussion mentions many features, but it would be helpful to have some sort of summary either as a table or specifically enumerated in the text.

Validity of the findings

The added discussion of Wu et al is helpful for understanding how Enhort can not only reproduce analyses but be used to rapidly extend investigations to develop and evaluate additional hypotheses. I appreciate the point raised by the response letter as to challenges in evaluating these kinds of systems. However, there are several points in the evaluation that would benefit from further detail. For example, what proportion of analyses in the literature review were conducted using new custom scripts versus other tools or replicating prior analyses (which would imply reuse)? While these details may not always be available, commenting on specific instances where something concrete can be said about the way the analyses were conducted (e.g., looking at provided supplements or methods mentioned in the paper) would significantly strengthen the claims made in this section. If no such examples can be provided, the speculation about time saving should be removed as the current literature review provides evidence of the utility of the target analyses but does not provide any evidence in support of time savings.

Table 1 is a beneficial addition to understand the processing power of Enhort. While a discussion of where the computational efforts are taking place is not necessary given the scope of the paper, it would be helpful to note the hardware these analyses were conducted on for reproducibility (either the client side specifications if client-side computation was conducted or the server specifications if all computation is conducted on the server).

In the re-analyses, it would be helpful to at least speculate on reasons why Enhort's results varied from the original analyses from Wilson et al. and Wu et al. The detailed expansion on the Wu et al. analyses provides a useful example of the utility of Enhort's approach and its importance for improved data analysis; however, the differences in the two outcomes may lead to some concerns around flexibility: does Enhort's automated workflow prevent scientists from tuning their analyses with sufficient precision? A discussion of where these variations might arise would be useful in understanding potential trade-offs.

Additional comments

Overall, the paper has been significantly improved. However, the manuscript should still provide a more explicit and grounded discussion of current practices to highlight the utility and trade-offs of Enhort.

---

## Round 0.3 · accepted · Accept

Thank you once again for addressing all the reviewers' comments. Only typographic and layout issues remain - please ensure these are addressed during the proof stage.

1. Figure 1 Legend: 'WebLab' should be "Wet lab"
2. Line 66. Please cite Circos' canonical publication when it is first mentioned: Krzywinski et al.
3. Line 90. 'A set of additional features listed in the following table'
* the table should be typeset as a table, rather than included in the text (please work with PeerJCS editorial staff to ensure this is done !)
* You probably mean to say 'A summary of Enhort's additional features are given in Table 1'.
4. Page 6 - table of analysis execution times. Numeric columns should be left justified. It may help to use scientific notation in order to more easily compare values.